# Fabrication and Tribological Properties of Copper Matrix Solid Self-Lubricant Composites Reinforced with Ni/NbSe_2_ Composites

**DOI:** 10.3390/ma12111854

**Published:** 2019-06-07

**Authors:** Fei-xia Zhang, Yan-qiu Chu, Chang-sheng Li

**Affiliations:** 1Key Laboratory of Tribology of Jiangsu Province, School of Materials Science and Engineering, Jiangsu University, Zhenjiang 212013, China; zhangfx@zjc.edu.cn; 2School of Mathematical, Jiangsu University of Science and Technology, Zhenjiang 212013, China; chuyanqiu@just.edu.cn; 3School of Modern Equipment Manufacturing, Zhenjiang College, Xuefu Road, Zhenjiang 212003, China

**Keywords:** copper-based composites, friction coefficient, tribological properties

## Abstract

This paper presents a facile and effective method for preparing Ni/NbSe_2_ composites in order to improve the wettability of NbSe_2_ and copper matrix, which is helpful in enhancing the friction-reducing and anti-wear properties of copper-based composites. The powder metallurgy (P/M) technique was used to fabricate copper-based composites with different weight fractions of Ni/NbSe_2_, and tribological properties of composites were evaluated by using a ball-on-disk friction-and-wear tester. Results indicated that tribological properties of copper-based composites were improved by the addition of Ni/NbSe_2_. In particular, copper-based composites containing 15 wt.% Ni/NbSe_2_ showed the lowest friction coefficient (0.16) and wear rate (4.1 × 10^−5^ mm^3^·N^−1^·m^−1^) among all composites.

## 1. Introduction 

Owing to its low cost, ease of production, and excellent thermal and electrical conductivities, copper and its composites have always attracted considerable interest among all metallic materials [1,2,3,4]. These merits have made copper-based composites become potential materials in industrial applications, such as for automobiles, marines, and machinery. However, especially in the field of tribology, the application of copper and its composites was seriously limited due to their low hardness and strength. Therefore, it is of vital significance to develop the friction-reducing and anti-wear properties of copper-based composites. So far, filling various suitable solid lubricants might be an effective method. Kumar and Kovacik et al. [5,6] found that copper matrix composites exhibit outstanding tribological properties compared with pure copper by the addition of MoS_2_ or graphite. However, MoS_2_ shows excellent lubricating properties under dry conditions, and graphite is a famous solid lubricant in a non-vacuum environment. Besides, both graphite and MoS_2_ could decrease load-carrying capacity and wear rate due to poor mechanical properties [7]. Therefore, the distinct shortcomings of MoS_2_ and graphite greatly limit the development of copper-based composites.

The exploration of effective solid lubricants is currently imperative. Transition metal dichalcogenides NbSe_2_ possesses a hexagonal laminated structure and outstanding self-lubrication ability, such as that of graphite and MoS_2_ [8]. Moreover, the mechanical property of NbSe_2_ is higher than that of graphite and MoS_2_, which could further improve the wear resistance of copper-based composites. Therefore, NbSe_2_ can be used as the substitute of MoS_2_ or graphite. Some researchers, such as Shi et al. [9] found that the friction-reducing and anti-wear properties of copper-based composites with nano-sized NbSe_2_ improved compared to those with micro-sized NbSe_2_, and the formation of tribo-film at the contact surface were strongly affected by the size of NbSe_2_ particles. Zheng et al. [10] found that the addition of NbSe_2_ could effectively reduce the plastic deformation of the substrate, making the friction and wear properties of the composite greatly improved.

As is well-known, the tribological of composites is also effectively enhanced by improving interfacial combination between the additives and matrix [11]. Additives covered by metal are the main method for obtaining a good interfacial combination. Chen et al. [12] reported that the addition of copper-coated CNTs and NbSe_2_ in the copper matrix meant their mechanical and tribological properties were much improved, compared with that of uncoated CTNs and NbSe_2_. Cui et al. [13] also found that Ni-coated graphite added into a bronze matrix can obtain a good interfacial combination, and make the wear resistance improve further. However, many investigations of Cu-based composites focus on tribological properties, and there are hardly any studies on copper-based composites with Ni-coated NbSe_2_ (Ni/NbSe_2_).

In this work, the electroless plating was used to deposit an Ni layer onto the surface of NbSe_2_, which effectively increased the interface combination between copper matrix and NbSe_2_. The density, hardness, and tribological properties of composites were not only investigated by adding Ni-coated NbSe_2_ to copper matrix, but the corresponding wear mechanisms are also discussed.

## 2. Experimental Procedures

### 2.1. Fabrication of Materials

Firstly, NbSe_2_ were successfully synthesized by a facile thermal solid-state reaction [14]; then, the as-prepared NbSe_2_ was coated with Ni layers by electroless plating [15]. Details of the electroless plating are as follows: NbSe_2_ powder was firstly coarsened by ultrasonic treatment in a concentrated H_2_SO_4_ solution for 20 min. Then, they were further sensitized and activated by immersion in an aqueous solution containing HCl (20 ml/L) and SnCl_2_ (20 g/L), and 0.3% AgNO_3_, respectively. The NbSe_2_ was immersed in copper plating solution for 30 min with agitation. (Table 1 exhibits the chemical components of the copper plating solution.) Finally, the copper-coated NbSe_2_ was dried in an oven at 100 °C for 12 h.

The mixed powders of copper and Ni/NbSe_2_ were ball-milled by a planetary high-energy mill machine (QM-SP2, NanDa Instrument Company, Nanjing, China) in a high-purity argon atmosphere for more than 600 min at a rotational speed of 250 rpm. The fraction of Ni/NbSe_2_ was selected to be 0, 5 wt.%, 10 wt.%, and 15 wt.% (designed as CNE0, CNE5, CNE10, and CNE15, respectively) for ascertaining the effect of the solid lubricant. Then, the above mixed powders were cold-pressed at 400 MPa and sintered pressurelessly at 800 °C in argon gas for 120 min. The samples were machined and polished before the properties test. For comparison, copper-based composite with 15 wt.% NbSe_2_ (denoted as CN15) was fabricated using the same processing parameters.

### 2.2. Characterization

The morphology of Ni/NbSe_2_ was characterized by a field emission scanning electron microscope (FESEM, JSM-7001F, JEOL, Tokyo, Japan) and TEM (Transmission electron microscope) (JEOL-2010, JEOL, Tokyo, Japan).

Micro-hardness was measured by a Vickers indenter (MH-5). 500 g and 15 s were a normal load and dwell time, respectively. The densities of composites were measured by using the Archimedes method. A ball-on-disk friction-and-wear tester (TRN) was used to evaluate the tribological properties of the samples. The disks and the counterpart were as-prepared materials and the GCr15 steel ball had a diameter of 4 mm, respectively. The surface of the disks and the counterpart were cleaned with acetone, and then dried in a vacuum before the wear test. Each test was conducted at the condition of 5N-0.00612 m/s for 20 min. The wear rate was expressed as the wear volume divided by the applied load and sliding distance [16]. The morphology and phase compositions on the wear scars of specimens were examined via a field emission scanning electron microscope (FESEM, JSM-7001F) equipped with EDS (Energy Dispersive Spectrometer). Three repeated wear tests were conducted on each sample, and the mean value was shown.

## 3. Results and Discussion

### 3.1. Microstructure of Composites

The morphological and structural characteristics of composites have been characterized in Figure 1. As shown in Figure 1a, NbSe_2_ had a mainly hexagonal structure, with the average diameter being about 2 µm and 200 nm in thickness. Figure 1b shows the morphology of Ni/NbSe_2_. The products consisted of a large number of Ni nanoparticles and sheet-like NbSe_2_. It can be clearly seen that Ni nanoparticles with average sizes of about 30 nm in diameter were well-dispersed and attached to NbSe_2_ sheets, which was consistent with TEM images in Figure 1d.

### 3.2. Characterization, Density, and Vickers Hardness

Figure 2 shows the microstructure of sintered CN15 and CNE15, which was characterized by optical micrographs after metallographic preparation. It can also be observed that the microstructure of specimen CNE15 was more homogeneous than that of specimen CN15. Figure 3 indicates the compactness of Cu/Ni/NbSe_2_ composites. It can clearly be seen that the compactness of specimen CN15 is lower than that of specimen CNE15. According to Figure 2 and Figure 3, it can be concluded that Ni-coated NbSe_2_ does not benefit the uniform dispersion of NbSe_2_ in the copper matrix, but further enhances the interfacial combination [16,17]. For specimen CN15, it can be seen that the NbSe_2_ phases have some agglomeration in the copper matrix, significantly deteriorating the physical and tribological properties of the composites [18].

Table 2 lists the compositions, densities, and hardness of composites. The densities of specimen CNE5, CNE10, CNE15, and CNE0 were 7.1 g·cm^−3^, 6.9 g·cm^−3^, 6.7 g·cm^−3^, and 7.3 g·cm^−3^, respectively. Compared with specimen CNE0, specimen CNE5, CNE10, and CNE15 showed lower densities for the addition of low-density NbSe_2_ [19]. Besides, the hardness of specimen CNE5, CNE10, CNE15, and CNE0 were 90 HV, 95 HV, 108 HV, and 75 HV, respectively. Specimen CNE5, CNE10, and CNE15 exhibited higher hardness than specimen CNE0 due to the dispersion strengthening effect of Ni/NbSe_2_. However, specimen CN15 possessed lower hardness than specimen CNE15. The presence of Ni with high hardness enhanced the mechanical properties of composites [20].

### 3.3. Tribological Properties

The variation of friction coefficients and wear rates of all five materials at the condition of 5N-0.00612m/s are illustrated in Figure 4. In Figure 4a, it can be seen that specimen CNE0 shows the highest and most unstable friction coefficient value of around 0.42, whereas with Ni/NbSe_2_ composites added, the composites exhibit a relatively low and stable friction coefficient value. In particular, composites show the lowest and most stable friction coefficient of about 0.16 as the Ni/NbSe_2_ increases to 15 wt.%. Besides, specimen CN15 shows a lower friction coefficient of about 0.27 than specimen CNE0, but it is higher than specimen CNE15.

As can be seen in Figure 4b, the wear rate of composites obviously decrease due to the addition of Ni/NbSe_2_. It can be clearly seen that the wear rate falls from 7.3 × 10^−3^ mm^3^·N^−1^·m^−1^ to the lowest value of 4.1 × 10^−5^ mm^3^·N^−1^·m^−1^ as the Ni/NbSe_2_ content increases to 15 wt.%. However, the wear rate of specimen CN15 is about 2.3 × 10^−3^ mm^3^·N^−1^·m^−1^, which is higher than that of specimen CNE15. Therefore, specimen CNE15 possessed the lowest wear rate, which decreased by more than 99% and 98% in comparison with specimens CNE0 and CN15, respectively. Thus, according to the above experimental results, Ni/NbSe_2_ showed better friction reduction and wear resistance than NbSe_2_ during the rubbing process.

Specimen CNE15 with the lowest friction coefficient and wear rate among all specimens was also proved by non-contact, three-dimensional cross-section images of wear tracks. As can be seen in Figure 5, the wear width of copper-based composites decrease with the addition of Ni/NbSe_2._ The wear width of specimen CNE5 is around 1.6 mm, which clearly decreased by 20% compared with that of specimen CNE0. In particular, specimen CNE15 possessed minimal wear width of about 0.45 mm, decreasing by 77%, 72%, and 52% compared with that of specimens CNE0, CNE5, and CN15, respectively.

Besides, the changing trend of wear width of composites was consistent with that of the wear depth. Therefore, the results were consistent with Figure 4b.

### 3.4. Worn Surface Morphologies

The worn surfaces of specimens CNE0, CNE5, CNE15, and CN15 after wear tests are shown in Figure 6. Figure 6a shows the worn surface morphologies of specimen CNE0. Furrows, plastic deformation, and clear adhesive took place on the worn surface of specimen CNE0, which was attributed to severe plastic deformation and adhesive wear [12,21]. This was consistent with the result of pure copper possessing the highest wear rate. In Figure 6b, grooves and wear debris can be observed on the worn surface of specimen CNE5, suggesting that the wear mechanism was dominant by micro-cutting and micro-ploughing, which meant that the addition of Ni/NbSe_2_ obviously changed the worn morphologies [10]. Besides, the grooves were indeed alleviated by the addition of Ni/NbSe_2_.

As can be seen in Figure 6c, there is slight plastic deformation and micro-scuffing on the worn surface of specimen CNE15. There is also the presence of a smooth, complete, and dense tribo-film, which helped to lower the friction coefficient and wear rate of specimen CNE15. However, the shape of the wear grove is not regular (Figure 5c) and its width and depth is not constant, indicating that the contact area and friction force oscillated and could have influenced the tribo-film.

Figure 6d exhibits the worn surface of specimen CN15. It can be seen that the worn surface is covered by the discontinuous tribo-film, which was responsible for increased friction coefficient of wear rate.

### 3.5. Phase Composition of Worn Surface

The corresponding EDS of the worn surface on the specimens is illustrated in Table 3. The elemental mapping revealed that Cu and O, as well as a small amount of Fe were distributed uniformly on the worn surface, showing that some oxides of copper and iron were formed on the worn surface of specimen CNE15 during the sliding process, and the Fe element was transferred from the counterpart of the GCr15 steel ball. This means that severe oxidative wear could be found on the worn surface of Cu-based composites during the sliding process.

Besides, the elements Ni, Nb, and Se were evenly distributed on the worn surface except for the elements of Cu, Fe, and O, indicating that the tribo-film and oxide layer were formed on the worn surface of specimen CNE15, which remarkably improved the tribological properties of copper-based composites [19]. Besides, as can be seen in Table 2, the content of the O element decreased with the addition of Ni/NbSe_2_ compared with specimen CNE0. It can thus be concluded that a massive amount of Ni were squeezed out of the matrix to spread out on the wear scar, which hindered the oxidation reaction rate and lowered the oxide forming rate, leading to the decrease in O content [22].

In Figure 7, the micro-Raman analyzer gives better insight into the local phase constitutes of the worn and unworn surfaces of specimen CNE15. By comparing the Raman spectrum of the tested and un-tested areas, NbSe_2_, Fe_2_O_3_, and CuO were found on the worn surface, and the intensity of NbSe_2_ was the strongest among all phases. In contrast, only relatively weak peak centering of about 230 cm^−1^ of NbSe_2_ was observed on the unworn surface. CuO was formed by the oxidation of metallic copper, and Fe_2_O_3_ was transferred from the counterpart GCr15 steel. The results were consistent with the EDS analysis.

Thus, it can be concluded that NbSe_2_ was the predominant phase in the worn surface, and the presence of NbSe_2_ was responsible for the improvement of tribological properties [23,24,25,26,27]. However, Ni could not be detected by the micro-Raman, meaning that it could not be observed on the worn surface.

The wear mechanism of composites with NbSe_2_ and Ni/NbSe_2_ are shown in Figure 8. For specimen CN15, NbSe_2_ squeezed out from the matrix due to the combined action of load extrusion and frictional heat were easily sheared to form the protective tribo-layer on the worn surface during the rubbing process, which avoided the direct contact between the substrate and friction pair. Besides, NbSe_2_ could continuously repair the tribo-layer under the load. However, the wettability between NbSe_2_ and substrate was poor, and some NbSe_2_ had a certain degree of agglomeration in the matrix, which led to a decrease in the mechanical properties of specimen CN15. Therefore, the formed tribo-layer was not tightly bonded to the substrate, and was easily broken and peeled off by the friction pair wear, resulting in serious wear of the composite.

For specimen CNE15, Ni/NbSe_2_ is tightly bound to the matrix and evenly distributed. During the rubbing process, Ni/NbSe_2_ was squeezed out from the matrix under the action of load extrusion and frictional heat. Due to the tight interface between the Ni-coated NbSe_2_ and copper matrix, the formed tribo-layer adhered tightly to the worn surface. The tribo-layer subjected to the shearing force of friction pair was easy to slip without breaking or falling off, thereby effectively improving the anti-friction and wear resistance of Cu-based composites. Besides, Ni nanoparticles with high hardness acting as micro-bearings could further decrease the friction coefficient and wear rate of composites. Therefore, the synergism of Ni and NbSe_2_ exhibited superior friction-reducing and anti-wear properties of Cu-based composites. Moreover, Ni/NbSe_2_ possessed better oxidation resistance than NbSe_2_, which could further reduce the oxidative wear of composites.

## 4. Conclusions

Ni/NbSe_2_ composites were successfully prepared by electroless plating. Ni nanoparticles with a particle size of 30 nm were uniformly coated on the surface of NbSe_2_.Copper-based composites exhibited lower friction coefficients and wear rates with the addition of Ni/NbSe_2_, which was attributed to the formation of the tribo-layer on the worn surface during the sliding process. Friction-reducing and anti-wear properties of Cu-based composites were improved significantly, which was attributed to a good interfacial combination between the NbSe_2_ and copper matrix, as well as the synergism of Ni and NbSe_2_. Moreover, Ni/NbSe_2_ possessed better oxidation resistance than NbSe_2_, which could further reduce the oxidative wear of composites.

## Figures and Tables

**Figure 1 materials-12-01854-f001:**
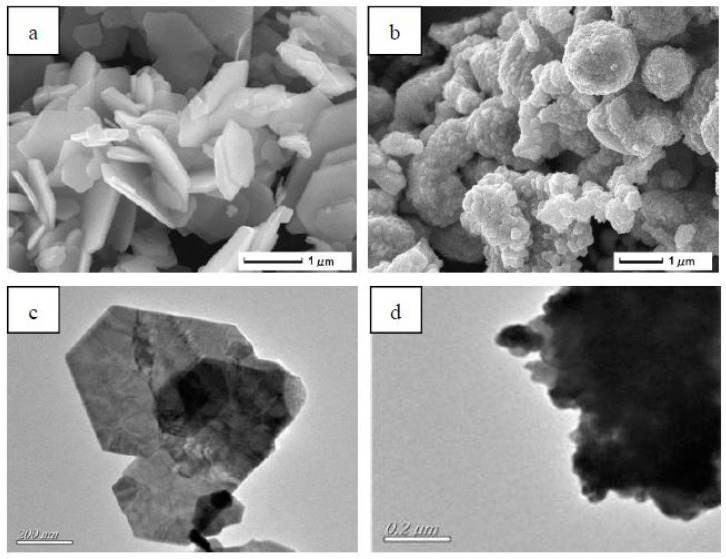
Characterization of NbSe_2_ and Ni/NbSe_2_ particles: (**a**) SEM of NbSe_2_, (**b**) SEM (scanning electron microscope) of Ni/NbSe_2_, and (**c**) TEM (Transmission electron microscope) of NbSe_2_, (**d**) TEM of Ni/NbSe_2._

**Figure 2 materials-12-01854-f002:**
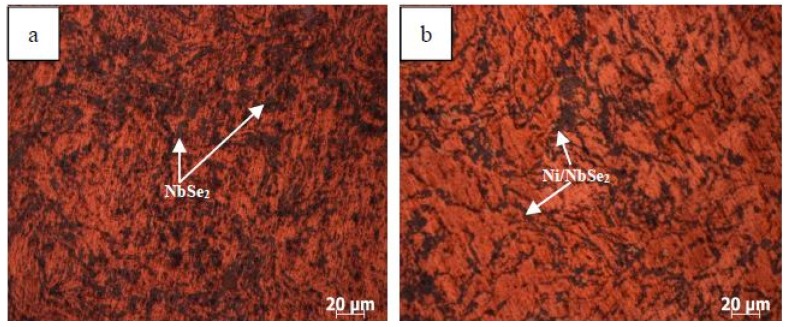
Microstructure of Cu/Ni/NbSe_2_ composites, CN15 (**a**) and CNE15 (**b**).

**Figure 3 materials-12-01854-f003:**
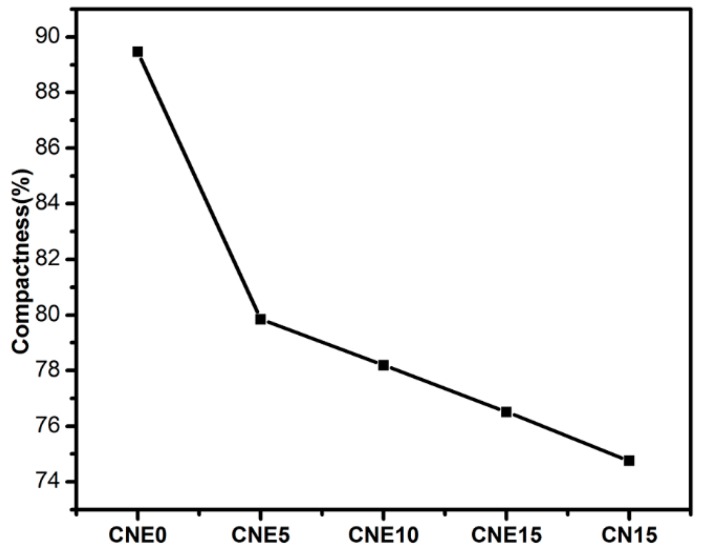
Compactness of Cu/Ni/NbSe_2_ composites.

**Figure 4 materials-12-01854-f004:**
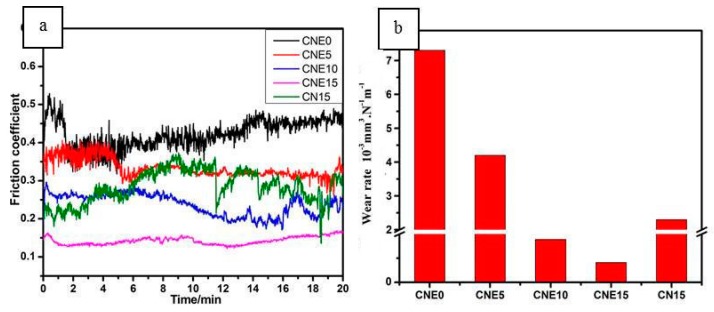
Friction coefficient (**a**) and wear rate (**b**) of Cu/Ni/NbSe_2_ composites.

**Figure 5 materials-12-01854-f005:**
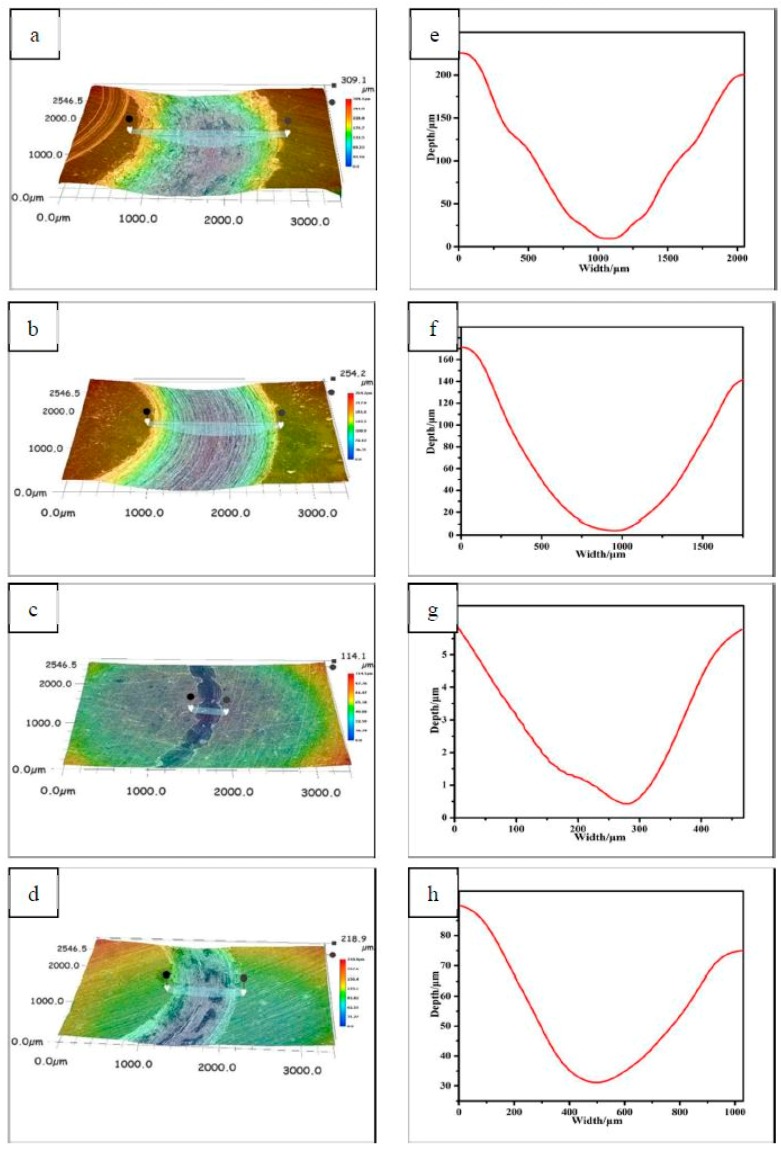
Non-contact, three-dimensional surface of (**a**) CNE0, (**b**) CNE5, (**c**) CNE15, and (**d**) CN15; (**e**–**h**) are the outline of the corresponding wear scar.

**Figure 6 materials-12-01854-f006:**
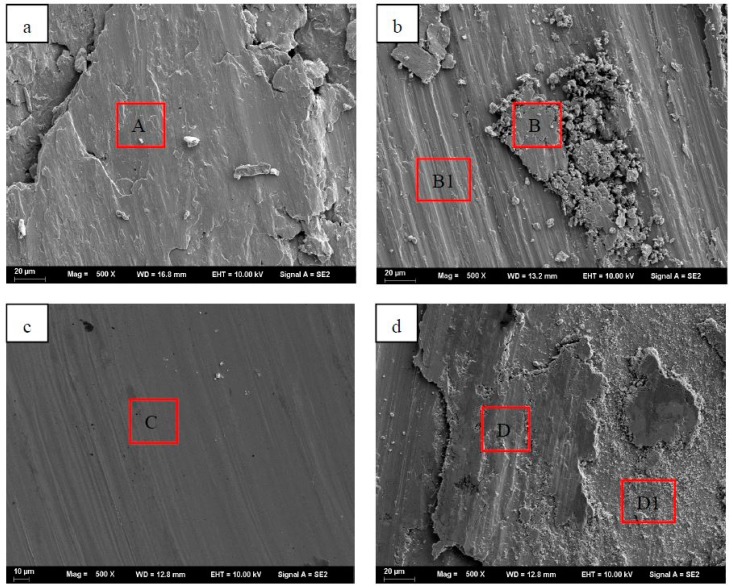
SEM image of worn surfaces of (**a**) CNE0, (**b**) CNE5, (**c**) CNE15, and (**d**) CN15.

**Figure 7 materials-12-01854-f007:**
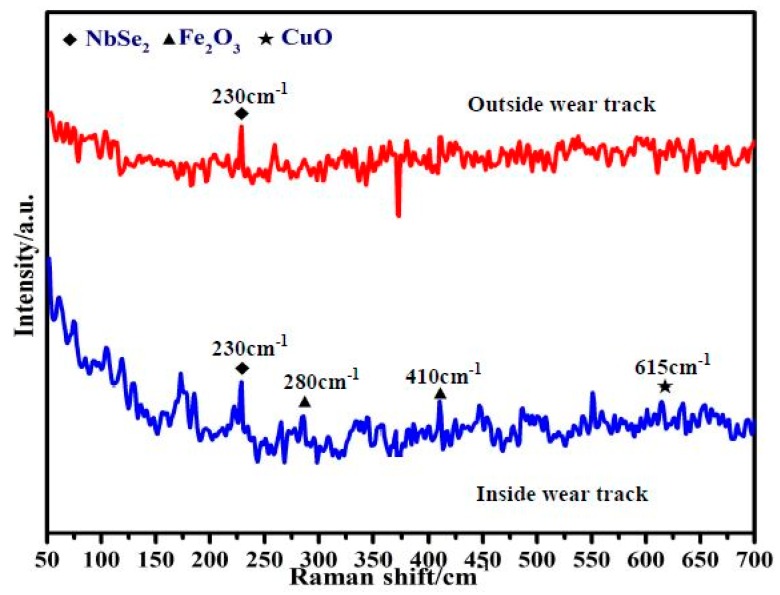
Raman spectrum of worn and unworn areas of specimen CNE15.

**Figure 8 materials-12-01854-f008:**
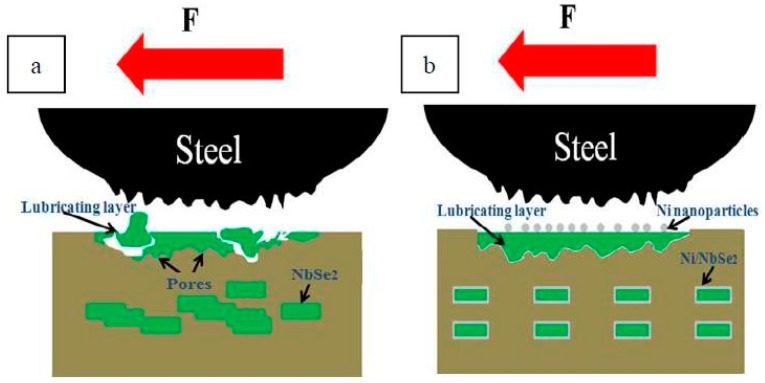
Anti-friction and wear-resistance mechanism of Cu-based composites (**a**) Cu/NbSe_2_, (**b**) Cu/Ni/NbSe_2._

**Table 1 materials-12-01854-t001:** Chemical composition of the copper plating solution.

Composition	Concentration(g/L)
NiSO_4_·5H_2_O	25
HCHO	20~40
EDTA-2Na	20~32
C_4_O_6_H_4_KNa	10~20
NaOH	10

**Table 2 materials-12-01854-t002:** Density and hardness of Cu/Ni/NbSe_2_ composites (mass fraction/wt.%).

Specimen	Cu	Ni/NbSe_2_	NbSe_2_	Density (g/cm^3^)	Hardness (HV)
CNE0	100	0	0	7.3	78 ± 4.1
CNE5	95	5	0	7.1	90 ± 5.8
CNE10	90	10	0	6.9	95 ± 4.9
CNE15	85	15	0	6.7	108 ± 6.0
CN15	85	0	15	6.4	101 ± 7.9

**Table 3 materials-12-01854-t003:** EDS of worn surface of Cu/Ni/NbSe_2_ composites (mass fraction/wt.%).

Area	Composition (wt.%)					
	Cu	Fe	Ni	Nb	Se	O
A	84.85	1.56	0	0	0	13.59
B	85.01	2.34	1.48	0.85	1.76	8.56
B1	90.2	1.25	0.9	0.43	1.03	6.19
C	78.08	4.88	3.80	2.73	4.08	6.43
D	61.89	0.53	0	3.83	7.20	26.55
D1	67.36	0.21	0	2.67	5.29	24.47

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
