# Peer review of "Fabrication and Tribological Properties of Copper Matrix Solid Self-Lubricant Composites Reinforced with Ni/NbSe2 Composites"

_materials, 2019, doi:10.3390/ma12111854_

Round 1

Reviewer 1 Report

In the paper the tribological properties of copper matrix composites reinforced with Ni/Nb Se2 are investigated. The results indicate that an addition  of  Ni/Nb Se2 to Cu matrix enables to achieve  an excellent wear resistance and low friction of the material. However there are some points that should be clarified.   

1.      The value of  wear rate. The value wear rate for pure copper (CNE0) measured by the Authors (730 x10-3 mm3.N-1m-1, Fig.4) is very far from the values reported in other papers, e.g. [*], where it is evaluated as 5.6×10-14 m3.N-1m-1=5.6 x10-5mm3.N-1m-1. Similarly the value of friction coefficient for pure Cu presented by the Authors is two times greater than that measured in [*].  It makes questionable other results presented in the paper. The Authors claim that due to the  Ni coated Nb Se2 (15%)  in the Cu matrix composite (CNE15),  the wear rate decreases of approximately 730/40=13 times, with respect to the pure copper, while in the reference [*], for a similar composite (denoted as C0), and fabricated with use of Cu coated Nb Se2 (15%),  we observe a decrease of wear rate  only of about 10-15% . So, the wear resistance achieved in the paper is rather unlikely and it should be clarified.  

2.      The explanation of the decrease of wear rate for the CNE15 composite. The Authors claim that  “slight plastic deformation and micro-scuffing were found on the worn surface of specimen CNE15. A smooth, complete and dense tribo-film were also noted...”, however the shape of wear grove is not regular (Fig. 5b),  its width and depth is not constant. It indicates that the contact area and friction force oscillate and  can influence the tribo-film. This point should be clarified.  

3.      In order to confirm the explanation of the increase of wear resistance presented in Fig. 8 (continuity or discontinuity of tribo-film) the EDS study should be shown for two different region of each worn surface (in Fig. 6 only one  region for each material is depicted)

4.      There are errors in the citation of the reference [12], some authors are neglected,  c.f. [*]

[*] B. Chen, J Yang, Q. Zhang, H. Huang, H. Li, H. Tang, C. Li Tribological properties of copper-based composites with copper coated  NbSe2 and CNT Materials and Design 75 (2015) 24–31

Author Response

Dear Reviewer:

Thank you for your comments concerning our manuscript entitled “Fabrication and tribological properties of copper matrix solid self-lubricant composites reinforced with Ni/NbSe2 composites(Manuscript ID: materials-477087). We have studied comments carefully and have made correction which we hope meet with approval. Revised portion are marked in blue in the revised manuscript. The main corrections in the paper according to the comments are as follows:

Point 1:The value of wear rate. The value wear rate for pure copper (CNE0) measured by the Authors (730 x10-3 mm3.N-1m-1, Fig.4) is very far from the values reported in other papers, e.g. [*], where it is evaluated as 5.6×10-14 m3.N-1m-1=5.6 x10-5mm3.N-1m-1. Similarly the value of friction coefficient for pure Cu presented by the Authors is two times greater than that measured in [*].  It makes questionable other results presented in the paper. The Authors claim that due to the Ni coated NbSe2 (15%) in the Cu matrix composite (CNE15),  the wear rate decreases of approximately 730/40=13 times, with respect to the pure copper, while in the reference [*], for a similar composite (denoted as C0), and fabricated with use of Cu coated NbSe2 (15%),  we observe a decrease of wear rate  only of about 10-15% . So, the wear resistance achieved in the paper is rather unlikely and it should be clarified.

Response 1: Thank you for your suggestions. However, in my paper, the applied load is 5N, which is far away from the applied load (1.5N) reported in other papers, e.g. [*]. High loads often can produce very large wear rate during the sliding process. Ni compared with Cu possess the higher hardnesss, thus, Ni can not only increases the interface combination between copper matrix and NbSe2, but also can significantly increase the hardness and wear resistance of composites. So, the wear resistance achieved in the paper is rather unlikely.

Point 2:The explanation of the decrease of wear rate for the CNE15 composite. The Authors claim that “slight plastic deformation and micro-scuffing were found on the worn surface of specimen CNE15. A smooth, complete and dense tribo-film were also noted...”, however the shape of wear groove is not regular (Fig. 5b), its width and depth is not constant. It indicates that the contact area and friction force oscillate and can influence the tribo-film. This point should be clarified.

Response 2: Thank you for your suggestions. Your advice has been clarified in the revised manuscript.

Point 3:In order to confirm the explanation of the increase of wear resistance presented in Fig. 8 (continuity or discontinuity of tribo-film) the EDS study should be shown for two different region of each worn surface (in Fig. 6 only one  region for each material is depicted).

Response 3: Thank you for your advice. It is really a mistake for us. The EDS study on two different region of worn surface of CNE5 and CN15(d) (in Fig. 6) was done, as indicated in Table 2. The EDS study on two different region of worn surface of CNE15 was futile, because the formed tribo-film covered the whole worn surface so that no other region could be found.

Point 4:There are errors in the citation of the reference [12], some authors are neglected,  c.f. [*]

Response 3:“B. Chen, J Yang. Li Tribological properties of copper-based composites with copper coated  NbSe2 and CNT[J]. Materials and Design 75, 2015, 24–31.” was corrected as B. Chen, J Yang, Q Zhang, H Huang, H Li, H Tang, C. Li. Tribological properties of copper-based composites with copper coated  NbSe2 and CNT[J]. Materials and Design 75, 2015, 24–31.”

Finally, we appreciate for your warm work earnestly, and hope that the correction above will meet with approval.

Once again, thank you very much for your comments and suggestions.

Yours sincerely

Changsheng Li

Reviewer 2 Report

With a mixture between Cu/Ni/NbSe2, the authors fabricated the composite and evaluated the tribological properties. The composites which show improved tribological properties were obtained and they show better oxidation resistance. The topic of this work is interesting and well within the aim of MATERALS journal. The submission can be accepted for publication after some revision.  

1) In Fig.1 c and d, authors should provide the SAED patterns obtained from the TEM images. It could be better for the reader’s understanding.

2) Authors mentioned that NbSe2 phases agglomerated in the copper matrix for CN15 specimen. It would be better to mark these in Fig. 3 for readability.

3) Could you please tender the hardness values of Cu 100%, Ni/NbSe2 100%, and NbSe2 100%, respectively? Authors can calculate the hardness values using the volume fraction and each hardness values and compare with the measured hardness. And then authors could say such advantage of the process what they mentioned. In addition, authors should provide the error range on all data.

4) On the 176~180 lines, authors mentioned the value of the wear rate. However, the units look like they need superscript. For example, 4.1x10-5mm3 à 4.1x10-5mm.

5) In Cu alloys and composites, it should be mentioned which values of the friction coefficient and the wear rate are the basis for the excellent properties.

6) Could you please mention the values of the tribological properties reported in other Cu-based composite and show that they are superior to others in comparison to the results mentioned in this manuscript?

7) The Raman data is unclear. Please give the high-resolution data and index it clearly.

8) At the introduction part, the last paragraph is very similar to the previous paper what they published (Materials & Design 75, 2015). Authors have to modify this.

9) Before final submission, authors have to check very carefully grammatical errors and spacing words.

Author Response

Responses to Reviewer

Dear Reviewer:

Thank you for your comments concerning our manuscript entitled “Fabrication and tribological properties of copper matrix solid self-lubricant composites reinforced with Ni/NbSe2 composites(Manuscript ID: materials-477087). We have studied comments carefully and have made correction which we hope meet with approval. Revised portion are marked in blue in the revised manuscript. The main corrections in the paper according to the comments are as follows:

Point 1: In Fig.1 c and d, authors should provide the SAED patterns obtained from the TEM images. It could be better for the reader’s understanding.

Response 1: Thank you for your suggestions. However, the SAED patterns needs to take much time to be obtained, which has not enough time for me. I sincerely hope to get your understanding.

Point 2: Authors mentioned that NbSe2 phases agglomerated in the copper matrix for CN15 specimen. It would be better to mark these in Fig. 3 for readability.

Response 2: It is really a mistake for me not to mark these in Fig. 3. This work has been done, as shown in the revised manuscript.

Point 3: Could you please tender the hardness values of Cu 100%, Ni/NbSe2 100%, and NbSe2 100%, respectively? Authors can calculate the hardness values using the volume fraction and each hardness values and compare with the measured hardness. And then authors could say such advantage of the process what they mentioned. In addition, authors should provide the error range on all data.

Response 3: Thank you for your suggestions. However, NbSe2 belongs to powders, which can not obtain its hardness. Thus, I can not calculate the hardness values using the volume fraction and each hardness values and compare with the measured hardness.

Point 4: On the 176~180 lines, authors mentioned the value of the wear rate. However, the units look like they need superscript. For example, 4.1x10-5mm3 à 4.1x10-5mm

Response 4: We have checked this sentence carefully, “4.1x10-5mm3” was corrected as 4.1x10-5 mm3.N-1m-1.

Point 5: In Cu alloys and composites, it should be mentioned which values of the friction coefficient and the wear rate are the basis for the excellent properties.

Response 5: We have checked this sentence carefully. For Fig.4a, specimen CNE0 showed the highest and most unstable friction coefficient among those of other specimens. Whereas with Ni/NbSe2 composites added, composites exhibited a relatively low and stable friction coefficient. Especially, composites showed the lowest and most stable friction coefficient of about 0.16, as the Ni/NbSe2 increased to 15wt.%. Besides, specimen CN15 showed lower friction coefficient value about 0.27 than specimen CNE0, but higher than specimen CNE15. was corrected as “For Fig.4a, specimen CNE0 showed the highest and most unstable friction coefficient value (around 0.42) among those of other specimens. Whereas with Ni/NbSe2 composites added, composites exhibited a relatively low and stable friction coefficient value. Especially, composites showed the lowest and most stable friction coefficient of about 0.16, as the Ni/NbSe2 increased to 15wt.%. Besides, specimen CN15 showed lower friction coefficient value about 0.27 than specimen CNE0, but higher than specimen CNE15.

Point 6: Could you please mention the values of the tribological properties reported in other Cu-based composite and show that they are superior to others in comparison to the results mentioned in this manuscript?

Response 6: In the study of chen et al, the friction coefficient and wear rate of Cu/NbSe2 composite sliding against stainless steel under the load of 5 N are about 0.27 and 4.1x10-5 mm3.N-1m-1. In the study of chen et al Kai et al, the friction coefficient and wear rate of Cu/NbSe2 composite sliding against stainless steel under the load of 5 N are 0.13 and 5.8×10-4mm3.N-1m-1. Compared with the above mention the values, Cu/Ni/NbSe2 composites possess the better tribological properties.

Point 7: The Raman data is unclear. Please give the high-resolution data and index it clearly.

Response 7: Thank you for your suggestions. However, the high-resolution data does not need be given. We have indexed it clearly, which can guarantee its readability.

Point 8: At the introduction part, the last paragraph is very similar to the previous paper what they published (Materials & Design 75, 2015). Authors have to modify this.

Response 8: It is really a mistake for me. “In this work, NbSe2 was preliminary coated with Ni layers by the means of electroless plating and then further to be used to enhance copper matrix. The effects of Ni coated NbSe2 on the density, hardness and tribological properties of composites had been investigated in detail, and the corresponding enhancement mechanisms had been proposed. Hopefully, this study can provide some guidance for designing high-performance copper-based self-lubricant composites.” was corrected as “ the electroless plating was used to deposit a Ni layer on the surface of NbSe2,which effectively increased the interface combination between copper matrix and NbSe2. The density, hardness and tribological properties of composites were not only investigated by adding Ni coated NbSe2 to copper matrix, but also the corresponding wear mechanisms had been further discussed.”

Point 9: Before final submission, authors have to check very carefully grammatical errors and spacing words.

Response 9: We have checked all sentences carefully. Thank you for your suggestions.

Finally, we appreciate for your warm work earnestly, and hope that the correction above will meet with approval.

Once again, thank you very much for your comments and suggestions.

Yours sincerely

Changsheng Li

Round 2

Reviewer 1 Report

Point 1 (c.f. previous review)  is still not clear. There are still discrepancies in the results presented by the Authors. In the corrected version, in the paragraph before Fig. 4, the maximum and minimum wear rates are estimated as 7.3 x10-3mm3/Nm (CNE0) and  4.1x10-5 mm3/Nm (CNE15) respectively, so the ratio of wear rates equals 178. In Fig. 4 in turn we have completely different values i.e. 730 x10-3mm3/Nm and 41 x10-3mm3/Nm respectively for CNE0 and CNE15, so the wear rates ratio equals 17.8. The Authors should decide which values are the correct. It should be noted that both values of ratio (178 or 17.8) are much higher than that given in [*], for material M (copper) and C0 (15% copper coated NibSe2), for which the ratio is approximately 5.6/4.5=1.24.  Thus, the Authors suggest that the Ni coating improve the wear resistance 17.8/1.24= 14.3 times or 178/1.24=143 times ! This large difference can not be explicated by different loads that were applied in the present paper and  in [*].  In the latter, it was shown that the increase of load from 1N to 5N leads to the increase of wear rate 1.8 to 4.2x10-5 mm3/Nm (C3 composite), so it is approximately 2.1 times.  

Other problem is wear resistance of pure copper evaluated by the Authors. In the revised paper it equals 7.3 x10-3mm3/Nm (in text before Fig. 4) or 730x10-3mm3/Nm=7.3x10-1mm3/Nm (in Fig. 4), while in [*] it equals 5.6x10-5 mm3/Nm, so the wear rate of copper reported in [*] is 7.3 x10-3/ 5.6x10-5=130 or 7.3 x10-1/ 5.6x10-5=13000 times lower than in the present paper. Here again, the difference in loads (1.5N and 5N) can not explain such a discrepancy.

Friction coefficient for copper. In the proposed paper we have lower load (1.5N) than in [*] (5N), but the friction coefficient (0.42) is about two times  greater than in [*] (0.21). However, in [*] an inverse relationship was shown (material C3), i.e. the friction coefficient increases when the load increases.  

These discrepancies make questionable other results presented in the paper. The improve of wear resistance is a fundamental achievement of the paper, so the presented wear and friction parameters should be reliable.   

Author Response

Responses to Reviewer

Dear Reviewer:

Thank you for your comments concerning our manuscript entitled “Fabrication and tribological properties of copper matrix solid self-lubricant composites reinforced with Ni/NbSe2 composites (Manuscript ID: Materials-477087). We have studied comments carefully and have made correction which we hope meet with approval. Revised portion are marked using the “Track Change” function in the revised manuscript. The main corrections in the paper according to the comments are as follows:

1. Point 1 (c.f. previous review) is still not clear. There are still discrepancies in the results presented by the Authors. In the corrected version, in the paragraph before Fig. 4, the maximum and minimum wear rates are estimated as 7.3x10-3mm3/Nm (CNE0) and 4.1x10-5 mm3/Nm (CNE15) respectively, so the ratio of wear rates equals 178. In Fig. 4 in turn we have completely different values i.e. 730x10-3mm3/Nm and 41 x10-3mm3/Nm respectively for CNE0 and CNE15, so the wear rates ratio equals 17.8. The Authors should decide which values are the correct. It should be noted that both values of ratio (178 or 17.8) are much higher than that given in [*], for material M (copper) and C0 (15% copper coated NbSe2), for which the ratio is approximately 5.6/4.5=1.24.  Thus, the Authors suggest that the Ni coating improve the wear resistance 17.8/1.24= 14.3 times or 178/1.24=143 times! This large difference cannot be explicated by different loads that were applied in the present paper and in [*].  In the latter, it was shown that the increase of load from 1N to 5N leads to the increase of wear rate 1.8 to 4.2x10-5 mm3/Nm (C3 composite), so it is approximately 2.1 times.  

Reply: We are really sorry. The maximum and minimum wear rates are estimated as 7.3x10-3mm3/Nm (CNE0) and 4.1x10-5 mm3/Nm (CNE15), respectively, so the ratio of wear rates equals 178, this is revised in the revised manuscript. This large difference can be explicated by different preparation methods of copper-based composites and different applied loads. Repressed sintering method is used in [*]. Repressed sintering method compared with the method by us can make samples possess lower porosity and higher compactness, which leads to the samples show higher hardness and the excellent wear resistance. Thus, copper made by repressed sintering method compared with copper made by the method by us has higher hardness and density, which is consistent with in [*]. In addition, higher loads often can produce the higher wear rate during the sliding process. The combination of two factors leads to this large difference.

2. Other problem is wear resistance of pure copper evaluated by the Authors. In the revised paper it equals 7.3 x10-3mm3/Nm (in text before Fig. 4) or 730x10-3mm3 /Nm= 7.3x10-1 mm3/Nm (in Fig. 4), while in [*] it equals 5.6x10-5 mm3/Nm, so the wear rate of copper reported in [*] is 7.3 x10-3/ 5.6x10-5=130 or 7.3 x10-1/ 5.6x10-5=13000 times lower than in the present paper. Here again, the difference in loads (1.5N and 5N) can not explain such a discrepancy

Reply: The wear rate of pure copper is 7.3x10-3mm3/Nm, which is an error in the drawing. The wear rate of copper reported in [*] is 7.3 x10-3/ 5.6x10-5=130, which is reasonable. Because repressed sintering method compared with the method by us can make samples possess lower porosity and higher compactness, which leads to the samples show higher hardness and the excellent wear resistance. Thus, copper in my paper compared with copper in [*] show lower hardness and high porosity, which can deduce that copper in my paper compared with copper in [*] has worse wear resistance. In addition, the difference in loads (1.5N and 5N) also plays a role.

3. Friction coefficient for copper. In the proposed paper we have lower load (1.5N) than in [*] (5N), but the friction coefficient (0.42) is about two times greater than in [*] (0.21). However, in [*] an inverse relationship was shown (material C3), i.e. the friction coefficient increases when the load increases.

Reply: In our paper, the friction coefficient of copper under the applied load of 5N rather than 1.5N is about 0.42, which is higher than that of copper under the applied load of 1.5N in [*] (0.21). This is consistent with the result of [*]. The friction coefficient increases when the load increases.  

Finally, we appreciate for your warm work earnestly, and hope that the correction above will meet with approval.

Once again, thank you very much for your comments and suggestions.

Yours sincerely

Feixia Zhang